# Challenges with Verifying Microbial Degradation of Polyethylene

**DOI:** 10.3390/polym12010123

**Published:** 2020-01-05

**Authors:** Zahra Montazer, Mohammad B. Habibi Najafi, David B. Levin

**Affiliations:** 1Department of Biosystems Engineering, University of Manitoba, Winnipeg, MB R3T 5V6, Canada; zahra.montazer@umanitoba.ca; 2Department of Food Science and Technology, Ferdowsi University of Mashhad (FUM), Mashhad 91779-48974, Iran; habibi@um.ac.ir

**Keywords:** low-density polyethylene, abiotic degradation, biodegradation, microbial degradation

## Abstract

Polyethylene (PE) is the most abundant synthetic, petroleum-based plastic materials produced globally, and one of the most resistant to biodegradation, resulting in massive accumulation in the environment. Although the microbial degradation of polyethylene has been reported, complete biodegradation of polyethylene has not been achieved, and rapid degradation of polyethylene under ambient conditions in the environment is still not feasible. Experiments reported in the literature suffer from a number of limitations, and conclusive evidence for the complete biodegradation of polyethylene by microorganisms has been elusive. These limitations include the lack of a working definition for the biodegradation of polyethylene that can lead to testable hypotheses, a non-uniform description of experimental conditions used, and variations in the type(s) of polyethylene used, leading to a profound limitation in our understanding of the processes and mechanisms involved in the microbial degradation of polyethylene. The objective of this review is to outline the challenges in polyethylene degradation experiments and clarify the parameters required to achieve polyethylene biodegradation. This review emphasizes the necessity of developing a biochemically-based definition for the biodegradation of polyethylene (and other synthetic plastics) to simplify the comparison of results of experiments focused for the microbial degradation of polyethylene.

## 1. Introduction

Five types of petroleum-based polymers are the most commonly used to make single-use plastic materials, namely low-density polyethylene (LDPE), high density polyethylene (HDPE), polypropylene (PP), polyvinyl chloride (PVC), and polyethylene terephthalate (PET). LDPE, mainly used to make plastic carry bags and food packaging materials, is the most abundant petroleum-polymer on earth, and represents up to 64% of single-use plastics that are discarded within a short period after use, resulting in massive and rapid accumulation in the environment [1,2]. Despite recycling and energy recovery efforts, the harmful impacts of virtually “non-biodegradable” LDPE waste accumulation in landfill and in the oceans are increasing [3,4,5,6]. There is mounting evidence that micro-plastics are now found everywhere on the planet, including snow in the arctic [7]. Hence, a suitable method for disposal that is eco-friendly must be found [1,2,8].

Unlike organic wastes discarded by humans, polyethylene (PE), and other petroleum-based plastics, are extremely recalcitrant to natural biodegradation processes. The scientific literature contains a considerable number of reports on the biodegradation of synthetic plastics, and on PE in particular. Thirteen review articles on microbial and physical biodegradation mechanisms and microorganisms involved have been published since 2008 (Table 1). Although many studies have reported microbial degradation of PE, significant degradation of PE wastes has not yet been achieved at real scales. The lack of a working definition for biodegradation for polyethylene that can lead to testable hypotheses has limited our ability to develop a biochemically-based understanding of the mechanisms and processes involved in PE degradation.

Early microbial biodegradation experiments attempted to demonstrate that microbial activity could result in changes in the physical characteristics of plastics, such as tensile strength, water uptake, and crystallinity [22]. Microbial biodegradation of plastics was first reviewed by Pirt (1980) [23]. A decade later, Albertsson and Karlsson (1990) reported a 0.2% weight loss of PE after 10 years [22]. Otake et al. (1995) surveyed changes on the surface of PE polymers that had been buried in soil for 10 to 32 years [24]. A high degree of degradation was observed for thin films of LDPE. Although areas of the PE films with severe deterioration were characterized by whitening with small holes, overall rate of degradation was very low, even after years of exposure to soil microbes.

Some scientists have surveyed the aerobic biodegradation of treated polyethylene and/or polyethylene modified by the addition of additives (“addivitated”) PE in simulated soil burial and mature compost [25,26], in natural aqueous environments in laboratory condition [27,28], or in different type of soil contain microbial consortia in real condition [29]. Others tested the biodegradation of LDPE in soil and identified the microorganisms involved [30]. Abrusci et al. [31] isolated microorganisms adsorbed on the surface of PE films buried in agricultural soil and then tested the biodegradability of thermal and photo degraded addiviated LDPE films by those organisms.

Microbial degradation assay experiments usually include isolation of microorganisms from different sources by use of conventional, culture-dependent methods to find best potential microbial power to degrade polymeric PE chain. Some researchers have isolated potential microorganisms from different type of soil (garden soil, forest soil, garbage soil, mangrove soil, soil containing agricultural PE films for soil mulching) [32,33,34,35,36]. Plastic debris, solid waste dumps sites, or landfill areas (municipal solid soil) [8,37,38,39,40,41,42], water [2,43], waste water or sewage sludge [44], oil contaminated soil [45,46], and even from Waxworm larvae [47] were the other sources for the isolation of high potential PE-degrading bacteria.

The culture method involved parameters such as same constant incubation temperature (usually 30 °C) and aerobic culture condition over 3 to 10 days [33,39]. In these experiments, a large number of bacteria were identified as belonging to a limited number of genera (Table 2), but not all of them were responsible for PE degradation. Following the initial isolation of the bacteria, the ability of individual isolates to utilize treated and/or untreated polyethylene was investigated in pure shake-flask cultures over various periods of times. These bacteria were mostly identified by the use of sequencing 16S ribosomal RNA genes after amplification by polymerase chain reaction (PCR). In the final step, biodegradation assays with PE-degrading bacteria on polyethylene particles or films was estimated by different methods and techniques discussed in Section 6.3.

Because of the great variety of PE materials used and the wide-range of culture conditions, comparisons of the various results of biodegradation are not meaningful. This underscores the need for standardized methods and protocols to systematically study the biodegradation of synthetic plastics. To clarify issues arising from reports of microbial biodegradation of PE that were unsuccessful, in terms of degradation of PE at real scale, we must define the differences between degradation and deterioration, and define what the biodegradation process involves. In the following sections, we describe factors that are effective for microbial degradation of PE, and how these factors resulted in reports with unreliable percentages of PE biodegradation. Subsequently we propose an accurate description of the biodegradation process that will enable the accurate interpretation of biodegradation results.

## 2. Abiotic Deterioration of PE

The complete process of biodegradation has been divided into four stages: biodeterioration, biofragmentation, bioassimilation, and mineralization. However, before microorganisms can begin to attack PE, they need access points in the PE structure to start fragmentation. Thus, initially, oxidation of PE polymers occurs through abiotic process, such as exposure to ultraviolet (UV) irradiation [59] in combination with heat [60] and/or chemicals in the environment [61], without the action of microbes.

That oxidation of PE, especially oxidation induced by UV-irradiation, is usually accompanied by thermal aging, is well-established and the mechanisms of polymer transformation have been well demonstrated [59,62,63]. Previous research has reported the exposure of PE to UV-light or oxidizing agents generates carbonyl-groups in the alkane chains of PE, which are subsequently further hydrolyzed by microorganisms that catabolize the shorter PE chain reaction products (fragmentation). In this mechanism, initially, UV-radiation is absorbed by the polymer chain, which leads to radical formation. Eventually, oxygen is absorbed and hydroperoxides are formed, resulting in the production of carbonyl groups (Figure 1). Additional exposure to UV-radiation causes the carbonyl groups to undergo Norrish Type I and/or Type II degradation. Also, photo-oxidation can be initiated by impurities or pro-oxidants. UV-degradation can also begin at locations of trace hydroperoxide or ketone groups, introduced during the manufacturing process or fabrication.

The oxidative degradation of polyolefins can be followed by measuring the level of carbonyl group adsorption by infra-red spectroscopy (IR). The measured carbonyl groups are usually expressed as a carbonyl index (C.I.), defined as the ratio of carbonyl and methylene absorbances, was used to express the concentration levels of carbonyl compounds measured by ATR-FTIR. The ratio of the absorbance of the carbonyl peak at 1714 cm^−1^ [64] and that of the methylene absorption band at 1435 cm^−1^ (CH_2_ scissoring peak) taken as an internal thickness band (CI = A1714/A1435). The formation of carbonyl groups is increased by photo-oxidation, but also by increasing stress even after storage in an abiotic environment. Functional groups that can be identified by FTIR analysis are shown in Table 3.

If Norrish Type I or Type II degradation (or both) occur, additional peaks are observed in the IR spectrum of the polymer. For example, a terminal double-bond appears at 905–915 cm^−1^, and it is also possible to trace ester formation. Norrish Type I cleavage yields a carbonyl radical that can react with an alkoxy radical on the PE chain. A peak appears at 1740 cm^−1^ in the IR spectrum if this ester formation occurs. The plot of 1640–1850 cm^−1^ range of carbonyl groups, as determined by the overlapping bands corresponding to acids (1710–1715 cm^−1^), ketones (1714 cm^−1^), aldehydes (1725 cm^−1^), ethers (1735 cm^−1^), and lactones (1780 cm^−1^) can reveal the presence of different oxidized products. Yamada-Onodera et al. [65], Gilan et al. [36], Hassan et al. [33], Yashchuck et al. [26], Abrusci et al. [61], and Vimala and Mathew [50] all report UV-light as the most applicable method of photo-oxidation in PE biodegradation experiments. Figure 1 shows degradation pathways of polyethylene and production of different carbonyl group.

## 3. Biodeterioration of PE

In addition to the abiotic deterioration of PE materials, some microorganisms can initiate the oxidation process on their own, via the process of “hydroperoxidation”. This has been termed “biodeterioration”. However, the question as to whether PE oxidized in this manner can be ultimately degraded by microorganisms still remains to be clarified [10]. In some studies of microbial degradation of PE, different pro-oxidation additives (prodegradants) have been incorporated to the structure of polyethylene products to make them “oxo-degradable”. PE polymers containing products that render them oxo-degradable are referred to as “addiviated” polymers. Materials used to make addiviated PE polymers oxo-degradable include polyunsaturated compounds, transition metals like iron, cobalt, manganese, and calcium [31,44,57], totally degradable plastic additives (TDPA) with different commercial names [25,26,27], natural polymers (e.g., starch, cellulose, or chitosan), food grade dyes [40,43], or synthetic polymers containing ester, hydroxyl or ether groups [29] that are prone to hydrolytic cleavage by microorganisms.

In some comparative studies of the microbial degradation of PE, the deterioration of crude and addiviated PE polymers is initiated by abiotic parameters like sun-light [40,50], heat [43,56,58], or both [35,57], as well as the addition of oxidizing chemical agents like nitric acid [33,51], as forms of PE pretreatment to render the plastic more susceptible to microbial degradation. The effects of these treatments on PE structure, and subsequently microbial degradation, were then investigated and compared with samples that were not pretreated.

During the process of deterioration, a transformation in the basic structure of PE leads to the formation of oxidized oligomers and modification of the polymer. Deterioration by physical, biological, or chemical agents makes the PE fragile and sensitive to further oxidation by enzymes secreted by the microorganisms. In this stage, the structure of PE changes, but there is no fragmentation of the polymer, or reduction in molecular structure. Overall, the deterioration phase is characterized by an increase in access points for enzymes secreted by microorganisms, and a reduction of mechanical or other physical properties of the polymer.

## 4. General Overview of Biodegradation Processes 

The biodegradation process usually includes biofragmentation of the PE polymers by secreted enzymes, followed by bioassimilation of small cleavage fragments (molar mass must be less than 500 g/mol) by the microorganisms [56,66]. Many of the species shown to degrade PE are also able to consume linear n-alkanes like paraffin (C_44_H_90_, M_w_ = 618). The linear paraffin molecules were found to be consumed by several microorganisms within 20 days [58,67].

Microbial oxidation of n-alkanes is well understood and hexadecane, whose basic chemical structure is identical to that of PE, has been employed as a model compound for the investigation of the PE biodegradation and the relevant genes [46]. The initial step involves hydroxylation of C-C bonds to generate primary or secondary alcohols, which are further oxidized to aldehydes or ketones, and then to carboxylic acids. Thus, microbial oxidation decreases the number of carbonyl-groups due to the formation of carboxylic acids. Carboxylated n-alkanes are analogous to fatty acids, which can be catabolized by bacteria via the β-oxidation system pathway (Figure 2). However, neither cleavage of C-C bonds within the backbone of PE polymers, nor the generation of long carbon chain carboxylic acids hydrolysis products have been reported [45,46,68,69,70].

Studies of the genetic mechanisms associated with PE degradation are extremely scarce. However, it has been reported that Alkane hydroxylases (AlkBs), enzymes involved in the alkane hydroxylase system pathway, are known to degrade linear alkanes and are the best known enzymes involved in PE degradation in β-oxidation pathway [45]. The key enzymes of interest in the alkane hydroxylase system are monoxygenases. The number and types of Alkane hydroxylases vary greatly in different bacteria, in which the induction condition and amount of goal carbon in the alkane chain are completely different [71].

The *P. aeruginosa* genome encodes two Alkane hydroxylases, *alk*B1 and *alk*B2, while the *Rhodococcus sp.* TMP2 genome encodes 5 Alkane hydroxylases (*alk*B1, *alk*B2, *alk*B3, *alk*B4, and *alk*B5) [72]. The Alkane hydroxylase system has been investigated studied best in *P. putida* GPo1, which expressed an Alkane hydroxylase that participates in the first step of the n-alkane oxidation pathway by hydroxylating of the terminal carbon [73]. Yoon et al. [46] have shown that AlkB of *Pseudomonas aeruginosa* strain E7 actively degraded low molar mass PE and played a central role in the mineralization of LMWPE into CO_2_ [46]. Also, AlkB cloned and expressed in *Pseudomonas* sp. E4 was active in the early stage of in LMWPE biodegradation, even in the absence of the other specific enzymes like rubredoxin and rubredoxin reductase. Laccase enzymes (phenol oxidases) expressed by *Rodococcus rubber* are multi-copper enzymes that have also been shown to play a major role in PE biodegradation [55].

## 5. Factors Involved in Microbial Degradation Experiments of PE

There are a number of factors in microbial degradation of PE polymers that have a significant effect on the outcome and results of PE biodegradation experiments. Unfortunately, these factors were very often not considered in the planning and design of experiments reported in the literature. Consequently, the data presented in these reports has been unreliable and inconclusive with respect to PE biodegradation. These factors are described as follows. 

### 5.1. Polyethylene Structure and Shape

Accessibility of enzymes secreted by the microorganisms to the PE carbon chain is an important factor in microbial degradation. The microstructure of all PE material consists simply of linear carbon chains held together by hydrogen bonds. However, according to different manufacturing processes and subsequently different physical arrangements of the linear chains, polyethylene polymers can have different densities and 3-dimensional (3-D) structures (Figure 3): low molecular weight polyethylene (LMWPE); linear low-density polyethylene (LLDPE); low-density polyethylene (LDPE); high-density polyethylene (HDPE).

In addition, PE usually has a semi crystalline structure. Crystallinity of LDPE is about 45%–65% depending on the process method. Amorphous sections of LDPE generally contain short branches (10–30 CH_3_ groups per 1000 C-atoms), consisting of one or more co-monomers, such as 1-butene, 1-hexene, and 1-octene. This branching system prevents the PE molecules from stacking close together; making the LDPE chains at the surface more accessible and the tertiary carbon atoms that are present at the branch sites are more susceptible to attack. Also, impurities are more likely to be incorporated into amorphous regions.

Thus, it is important that the structure and degree of amorphous and crystalline regions in the polymer are reported, so that it is possible to know how much of the polymer chain is accessible to enzymatic attack. This additional information about the polymer used in PE biodegradation experiments will greatly support the correct interpretation of the results and comparison with data from similar experiments reported by other researchers.

In addition to LDPE, high density polyethylene (HDPE) is also a very common non-degradable petro-plastic waste [74]. LDPE degrades faster than HDPE, possibly due to the fact that the polymer chains of LDPE are more closely packed than those of HDPE and that LDPE has a lower content of vinylidene defects, which have been shown to be directly correlated with oxidization of the polymer. Further, there are fewer tertiary carbons in HPDE and its molar mass is much higher, possibly making it more difficult for microorganisms and/or their oxidizing enzymes to access the polymer chains [43,57]. Fontella et al. [57] compared the biodegradability of various pre-treated polyethylene materials, including HDPE films, LDPE films, and linear low-density polyethylene (LLDPE) films of different thicknessds, and containing pro-oxidant additives, by *Rhodococcus rhodochrous* (one of the most efficient bacteria for PE biodegradation). Biodegradation and microbial growth were measured by ATP and ADP assays, scanning electron microscopy, FTIR measurement, size exclusion chromatography, and NMR spectroscopy. Although all samples, except cobalt containing samples, were degraded by *R. rhodochrous*, HDPE was the least degraded and mineralization reached less than 6% after 317 days of incubation. This value is negligible and can be related to the consumption of compounds that were more easily degraded and consumed [57].

Devi et al. [74] studied biodegradation of HDPE by *Aspergillus* spp., based on weight loss and FTIR spectrophotometric analysis. The biodegradation of HDPE films was further investigated through SEM analysis. Sudhakar et al. [43] assessed biodegradation of unpretreated and thermally pretreated low-and high-density polyethylenes by *Bacillus* ssp. for one year. Biodegradation was evaluated by FTIR spectroscopy analysis and weight loss percentage. Degradation of un-treated pure samples according to wight loss percentage were 10% and 3.5% in cases of LDPE and HDPE, respectively. The ability of fungal isolates was proved to utilize virgin polyethylene as the carbon source without any pre-treatment and pro-oxidant additives [74]. In addition, basic polyethylene structure, physical properties, like density, and thermal properties, such as melting point, have also been reported. Mehmood et al. [40] and Hassan et al. [33] have reported the use of LDPE with melting points of 115 °C and 109 °C and densities of 0.93 g/cm^3^ and 0.921 g/cm^3^, respectively. This information can also help define similarities among PE polymers used in different biodegradation experiments.

Two other important factors, and possibly the two key factors, that should be reported in PE biodegradation experiments are Weight-averaged Molar mass (M_w_) and Number-average Molar mass (M_n_) (Polymers are all long carbon chains, but the lengths may vary by thousands of monomer units. Because of this, polymer molecular weights are usually given as averages. Two experimentally determined values are common: Mw, the weight average Molar mass, is the total weight of the polymer divided by the number of molecules of polymer in the sample; Mn, the number average Molar mass, is calculated from the mole fraction distribution of different sized molecules in a sample.). The main polymer chain usually has a M_w_ of more than 30,000 g/mol [25]. M_w_ is important because it indicates the required enzymatic power required to degrade the longest polymeric chain. Only a few reports in the literature actually indicate the M_w_ of the polymers tested. Polyethylene polymers of 4000 to 28,000 g/mol were investigated by Yamada-Onodera et al. [65], while Yoon et al. [46] tested the biodegradation of non-oxidized LMWPE of 1700 to 23,700 g/mol. These PE polymers were degraded by isolated *Pseudomonas sp.* E4 based on amount of CO_2_ evolved from the compost or percent of mineralization [46]. Abrusci et al. (2011) reported the M_w_, M_n_, and Polydispersity (PD) of original and accelerated photo-degraded polyethylene in their work [31]. Jeon and Kim (2014) reported Polyethylene powder with M_w_ of 1700 (LMWPE) was added to the enrichment medium of isolation and later extruded to form pellets used in their biodegradation experiment [44]. Chiellini and et al. (2003) reported thermally fragmented low-density polyethylene (LDPE) film samples containing totally degradable plastic additives pro-oxidants with M_w_ of 6720 were used in cylindrical glass vessels for biodegradation test [25]. Gilan et al. (2004) have used the branched LDPE film with a M_w_ of 191,000 [36].

Some researchers have used polyethylene powder in synthetic media, or enriched media to screen for, and isolate, the microorganisms active in PE biodegradation assays [33,38,39,42,52]. In the process of making polyethylene powder, usually a solvent like xylene is used to cleave the polyethylene structure while boiling, and then it is crushed and heated to 60 °C to evaporate the solvent and make recrystallized polyethylene powder or pieces. When the researcher uses solvents or liquid nitrogen to apply physical damage [37], in order to prepare polyethylene powder, fragmentation of the PE structure can occur, resulting in changes in the M_w_ and M_n_. As fragmentation happens, M_w_ or M_n_, and impurities in the PE powder used in the experiments should be measured and reported.

PE properties like M_w_ or melting point are usually noted in published reports, but only before other treatments like UV-irradiation, freezing in liquid nitrogen, extruding, or dissolving in solvents. As the length of polymer chain may be changed due to the treatment before exposure to microbial degradation, changes in M_w_, or M_n_ may be misinterpreted as solely due to microbial enzyme activity. In order to access the real PE-degradation power of microorganisms, the properties of PE immediately before it is exposed to microbial degradation must be reported.

### 5.2. Modification of Polyethylene

As described above, polyethylene and/or “degradable polyethylene” (Degradable PE(s) are types of polyethylene that contain some material like starch or polyvalent ions in order to accelerate the degradation in the environment. Degradable PE is also sometimes used when an polyester is produced with very few ester groups and properties similar to PE [75] can be treated by exposure to UV-light, heat, chemical oxidizing agents, liquid nitrogen, and/or chemical solvents to achieve different goals, such as sterilization of the PE surface, or making the powder preparation easy. However the main object of LDPE modification is to induce deterioration of the polyethylene structure, which allows greater access of enzymes secreted by microorganisms during the biodegradation stage. Meanwhile, treatments affect the structure of PE, so experiments have used different types of PE in terms of M_w_, M_n_, and/or molecular distribution have led to different biodegradation results. These changes should be determined and reported in the biodegradation process in order to accurately assess microbial degradation and to determine the pure effect of microorganisms’ activities.

For example, blending PE polymers with natural polymers like starch can increase their biodegradability. Karimi and Bitia [76] demonstrated that the enzyme α-amylase was able to degrade LDPE-starch blend samples in an aqueous solution, with the weight and tensile strength of the polymer samples reduced by 48% and 87%, respectively, after enzyme treatment. Gel permeation chromatography (GPC) revealed a significant reduction in both the molar mass and viscosity of the LDPE of more than 70% and 60%, respectively. The data from these experiments indicated that the main backbone of the polymer, as well as the side branches, have been cleaved by the enzyme, suggesting that α-amylase has a promiscuous co-metabolic effect on biodegradation of LDPE in polymer-starch blends.

While most of the early studies of microbial polyethylene degradation used PE that had been subjected to some form of pretreatment, several recent studies investigated microbial degradation of untreated polyethylene. Kyaw et al. [53] found that untreated LDPE could be degraded by four different strains of *Pseudomonas* including *P. aeruginosa* PAO1 (ATCC 15729), *P. aeruginosa,* (ATCC 15692), *P. putida* KT2440 (ATCC 47054), and *P. syringae* DC3000 (ATCC10862). After 120 days, the percentage of LDPE weight reduction was 20% in the *P. aeruginosa* PAO1 culture, 11% in the *P. aeruginosa* culture, 9% in the *P. putida* culture, and 11.3% in the *P. syringae* culture. Yoon et al. [46] showed that *Pseudomonas sp.* E4 was able to degrade non-oxidized LMWPE that had an average molar mass (M_w_) of 1700 to 23,700 g/mol. After microbial treatment, the surface of the LMWPE sheet was greatly deteriorated and eroded as a result of the microbial activity. More recently, Peixoto et al. [8] isolated bacteria in the genera *Cocomonas, Delftia,* and *Stentrophomonas* from plastic debris found in soil of the savanna-like Brazilian Cerrado and demonstrated that all strains were capable of degrading ultra-high molar mass polyethylene with molar mass up to 191,000 g/mol without additives or pre-treatments. Raman spectroscopy was used to confirm a significant loss in PE crystalline content.

### 5.3. Partial Biodegradation versus Complete Degradation

Complete biodegradation of PE polymers could be defined as the consumption and mineralization of intact, pristine polymers, including the backbone of the polymer. Yoon et al. [46] have commented that microbes that can completely degrade and mineralize pristine polyethylene have not yet been isolated. However, the many reports of PE biodegradation in the literature may be considered as partial biodegradation. As described above, PE polymers consist of a complex of linear carbon chains held together by van-der-Waals interactions, with accessible short side-chains with tertiary carbon that contain amorphous sections, terminal methyl-groups at the ends of chains, short branches, and small oxidative products, as well as many linear and branched n-alkane side-chains. The side-chains of PE resemble linear n-alkanes, and may be the first access point of enzymes secreted by bacteria that result in partial degradation of the polymers. Low molar mass molecules and/or amorphous segments are removed from the surface of the polymer without resulting in fragmentation of the backbone of the polymer [56,77]. Thus, weight loss in the early stages of PE degradation may be explained by enzymatic hydrolysis of these easily accessible side chains rather than fragmentation of the backbone or pristine PE polymers. The growth of microorganisms on agar plates containing polyethylene is not sufficient evidence for complete polyethylene biodegradation. Complete biodegradation must be confirmed, and this has been one of the major problems with biodegradation experiments.

Fragmentation of the intact polymer can only be established by measuring the M_w_ and M_n_. Small increases in M_w_ after microbial degradation indicate the consumption of low molar mass fragments, presumably the amorphous regions of PE [78]. In contrast, Yamada-Onodera et al. [65] reported a small decrease in the molar mass of oxidized (a combination of thermo-oxidation and chemical oxidation) polyethylene compared with that of unoxidized polyethylene, both of which had been incubated with *Penicillium simplicissimum* for three months. Yamada-Onodera et al. [65] exposed PE to UV-irradiation for 500 h followed by exposure to nitric acid for six days at 80 °C. The treated PE was then dissolved in xylene and re-crystallized before being used. The authors have not stated what was M_w_ after treatment (before fungal incubation), but it appears that this treatment was sufficiently severe to greatly increase the Carbonyl Index and facilitate oxidization by fungal enzymes, fragmenting the polymer chain and decreasing the M_w_.

Eyheraguibel et al. (2017) assessed the ability of *Rhodococcus rhodochrous* to biodegrade HDPE, and found that degradation and mineralization of the PE was a function of molecule size and oxidation state. They found that a large proportion of the extracted hydrolysis products had M_w_ lower than 850 g/mol, and that the maximum chain length of these oligomers was 55 carbon atoms. The oligomers were divided into chemically related compounds with different oxidation states ranging from 0 to 10, and 95% of the soluble oligomers were consumed after 240 days of incubation. Large, highly oxidized molecules were completely eliminated by the end of the experiment. Molecules containing one oxygen atom or less were less degraded, as were smaller molecules (<450 g/mol, 25 carbon atoms), suggesting that longer molecules were degraded and disappeared more rapidly than the smaller ones. This work provides a new insight into the microbial biodegradation processes, suggesting that extracellular mechanisms leading to chain cleavage may play a significant role in polyethylene biodegradation [79].

### 5.4. Interference of Other Carbon Sources in Biodegradation

There are some carbon sources that usually are consumed by bacteria in the early stages of microbial degradation in biodegradation experiments and may interfere with the sole carbon source of PE. To overcome the problem, it is recommended establishing a growth curve with PE as the carbon source for the bacteria under investigation. Changes in the growth curve may indicate consumption of different groups of carbon sources, with different accessibility by microorganisms [37]. Impurities incorporated to PE chain, or adhering to the PE surface, may consist of compounds that can be utilized as a carbon source by bacteria. Consumption of these contamination carbon sources can compete, or interfere, with consumption of PE as a carbon source. One way to reduce this problem is to incubate non-PE degrading bacteria, like *E. coli* with the impure PE samples. The *E. coli* would consume the impurities without altering the PE structure. After number of days, the purified PE could be retrieved, washed, and incubated with microorganisms of interest, to assess their true PE biodegradation potential. The following section discusses some carbon sources that could interfere with assessment of a microorganism’s true biodegradation potential.

#### 5.4.1. Carbon Sources Incorporated in Main PE Chain

One problem that can be seen in biodegradation assays with treated PE samples is that the bacteria have consumed only oxidative products of pretreated polyethylene, as sole carbon source, without breaking down the linear carbon chains of polyethylene. Some structural variations, such as unsaturated carbon-carbon double bonds, conjugated double bonds, vinyl groups, carbonyl groups, and hydroperoxide groups, formed during polymerization and subsequent processing, may also be present in the PE polymers that be used by bacteria [10,80].

The existence of these functional groups, as intentional or unintentional additives in polymer structure, will affect microbial accessibility of polymer. In the early stage of biodegradation, these segments may be consumed as carbon sources by bacteria and thus, may facilitate degradation the main polymer chain. As many reports have demonstrated, the use of UV-radiation and/or pro-oxidants can induce substantial oxidation and significantly improve biodegradation of polyethylene plastics the materials, only a few of the many papers published actually provide direct evidence of microbial biodegradation [25,27,31]. After degradation of the non-polymeric fractions, the remaining backbone of the polymer may disperse in the environment, but does not continue to degrade at any appreciable rate regardless of the environmental conditions, and may have separate downstream environmental impacts [14].

#### 5.4.2. Accidental Impurities Carbon Sources in PE Chain

Polyethylene products are often not pure and may be contaminated with other carbon compounds that may be consumed by bacteria. These impurities may include short carbon chains (oligomers and monomers) and other chemical materials that enter polyethylene beads during processing, and may be incorporated into amorphous regions of the polymer. These impurities may be consumed first by the bacteria, accounting for the rapid growth and consumption of intact polymer.

To overcome such a problem, Nanda and Sahu [41] washed PE pieces with ethanol to remove any organic matter adhering to the surface, then rinsed the PE Pieces in distilled water, and then air-dried them. The washed PE was crushed by grinding in a mortar pestle along with sufficient amount of crystalline NaCl to obtain a powder consisting of fine ruptured threads. The mixture was transferred into a conical flask with distilled water and mixed well in a shaker for 1 h. Crystalline NaCl was chosen for the purpose because the crystals would help in grinding and rupturing the polyethylene and its solution would wash away all impurities and organic matter adhering to it. The solution was passed through Whatman No. 41 filter paper. Polyethylene particles were recovered from over the filter paper gently and air-dried for the rest of experiments.

#### 5.4.3. Carbon Sources from Culture-Independent Methods

A significant problem when studying microbial degradation of polyethylene is that the polymer is solid and highly hydrophobic. To make this substrate accessible to microbes in an aqueous environment, liquid hydrocarbons such as paraffin oil or n-hexadecane have been added to preadapt the bacteria [2]. Other impurities that may interfere in microbial degradation of polyethylene are the materials added to microbial media in order to enhance dispensability of polyethylene. These include biosurfactant agents like EDTA, mineral oil, Triton-X100, and different Tweens (Tween 60 and Tween 80) [81]. Other organic matter adhering to the surface of PE particles also can interfere the sole carbon source consumption in PE degradation experiments. For example, solvents like Xylene added in order to solubilize PE powder may enter in the polymer structure as another carbon source, and be consumed by microorganisms, thus confounding the interpretation of the biodegradation experiment results.

Sub-culturing the inoculums from another culture in PE media may transfer some carbon material that may interfere in rest of the experiment. It should be considered that during culturing on PE media containing impurities, debris or biomass can be formed in early days of experiment that can be another source of carbon for consumption for new cells rather than PE in next days. The presence of two carbon sources, which may (or may not) compete as substrate may make it difficult to interpret microbial PE degradation data accurately.

## 6. Type of Microorganisms Used

Several microorganisms have been shown to grow on the surface of PE materials, among which are the species of the genus *Rhodococcus*, suggesting a potent ability of these microorganisms to use, at least partly, PE as a carbon source (Table 2). However, most *Rhodococcus* species, if not all, fail to induce a clear-cut degradation of PE samples [82]. A number of recent papers have identified, isolated, and chararcterized PE degradation by marine bacteria [83,84], including *Alcanivorax borkumensis* [85]. The great variation in results of PE degradation experiments reflects the great diversity of microorganisms used. Below, we describe the different ways in which microorganisms are applied in PE degradation experiments.

### 6.1. Polyethylene Degradation by Bacterial Consortia

A major advantage of using a pure culture of bacteria in biodegradation assays is the ability to distinguish between chemical degradation versus biological degradation [58]. The ability of individual bacteria isolates to degrade PE has been assessed using *Bacillus subtilis* [50], *Bacillus amyloliquefaciens* [39], *Acinetobacter buammi* [42], *Streptomyces* species [58], and *Rhodococcus ruber* [36]. Complex microbial communities [63] and combinations of selected bacteria have also been investigated to determine if there are synergistic effects on biodegradation by specific bacteria.

The rationale for using microbial consortia is that different microorganisms use different metabolic pathways and express different oxidative enzymes when cultured with the different plastic materials, and this combination should enable more effective microbial degradation. Abrusci et al. [31] compared the effect of a mixture of isolated *Bacillus* species and *Brevibacillus borstelensis* (DSMZ 6347) on oxo-biodegradable polyethylene, over 90 days at different temperatures. Biodegradation of the polyethylene was more effective when *B. borstelensis* (DSM-No 6347) used, at higher temperature (45 °C), with materials that had been exposed to light in the 300–800 nm range for over 500 h, containing Ca- and Fe-Stearates. The results of this experiment showed that application of consortia does not necessarily lead to higher biodegradation.

### 6.2. Fungi versus Bacteria in Biodegradation of Polyethylene

In addition to many studies of biodegradation of plastic by bacteria, the ability of fungi to attack polyethylene has been investigated (Table 4). Fungi are a rich source of oxidative enzymes and have the ability to survive in harsh environments under low nutrient and moisture conditions. In addition, they have the ability to extend hyphae that can penetrate into cracks and crevices [86]. Thus, fungi have great potential for the biodegradation of PE and other synthetic plastics.

Yamada-Onodera [65] demonstrated that *Penecillium simplicissimum* could grow on LDPE as sole carbon source. Kawai et al. [77] compared polyethylene biodegradation by consortium of soil bacteria and *Aspergillus sp.* using mathematical models based on the gradual weight loss of high molar mass molecules via the β-oxidation pathway versus direct consumption of low molar mass molecules by the cells. They concluded that bacterial β-oxidation of PE was 36-fold more effective than fungal β-oxidation. Kawai et al. [77] demonstrated that PE degradation by a bacterial consortium was effective for high molar mass (M_w_ = 5000 g/mol) polymers, while fungal degradation was ineffective when the polymer M_w_ was 1600 g/mol or less, and the lower limit in size for direct consumption by cells was estimated to be approximately 1500 g/mol.

Based on the reported literature, there are some limitations to the ability of fungi to degrade polyethylene: (1) Fungal hyphae grow on the surface of plastic materials and hydrolyze the upper layers rather penetrate into the backbone structure, while bacteria penetrate the polymer chains by the secretion of oxidative enzymes that are able to degrade the lower layers of the material; and (2) the formation of fungal mats on the surface of the PE could insulate the cells closest to the surface of the material from macro-and/or micro-nutrients, and/or oxygen. This insulating effect may explain some of the variation in biodegradation of plastics that has been observed with fungi [58].

### 6.3. Using Bacteria that Can Form Biofilms and Secrete Biosurfactants

Overall, surfaces of materials that are in contact with water are readily colonized by microorganisms, which form biofilms. Colonization and biofilm formation on PE have been well studied [36,39,56]. The bacteria most frequently reported to form biofilms on the surface of polyethylene belong to the genera *Pseudomonas* [41], *Rhodococcus* [36,90], and *Bacillus* [39]. These species have been shown to have the greatest potential for polymer degradation. In the case that polyethylene films are used in degradation experiment, the thickness and access area of layers will affect penetration of microorganisms in the structure. Also, characteristics of the surface will have effects on biofilm formation, especially when biofilm forming or surfactant producing microorganisms are used for microbial inoculation, hence notification of thickness and hydrophobicity of films are necessary in polyethylene film biodegradation. Different works have reported different thickness and dimension of polyethylene films (Table 2).

Biofilm formation has been assessed by different methods, including biofilm quantification by direct counting of the number of cells adhering to the surface and/or protein concentration [39]. Further, contact angle measurements of standard testing liquids and the subsequent calculation of surface tension components [35] have used to evaluate of hydrophobicity of material. Cell surface hydrophobicity or bacterial adherence to hydrocarbon (BATH) tests also was used by Das and Kumar (2013).

An important factor in biofilm formation has been attributed to cell surface hydrophobicity [39]. However, Eubeler et al. [68] demonstrated that biofilm formation does not necessarily result in biodegradation. Many microorganisms can form biofilms and do not degrade the material on which they live. Kounty et al. [35] isolated twelve strains that adsorbed to, and grew on, the surface of oxidized PE films containing pro-oxidant compounds. The majority of isolates belonged to the phylum Rhodococcus, and phylum Actinobacteria were also frequently isolated. The majority of these PE-degrading bacteria did not exhibit significant cell surface hydrophobicity. Thus, biofilm formation and PE-degradation are not necessarily directly linked to surface hydrophobicity. In contrast, secretion of biosurfactant molecules by bacteria appears to play an important role in colonization and degradation of PE. For example, surfactin is a biosurfactant compound produced by *Bacillus subtilis* [50].

## 7. Experimental Conditions

Experimental conditions, like incubation time, media type, temperature, and aeration, determine which microorganisms can grow on PE containing media. As conditions often are not consistent within biodegradation experiments, the biodegradation values reported for these experiments differ greatly, invalidating comparisons of the biodegradation results. Polyethylene degrading bacteria are often screened, in minimal salts culture medium containing plastic powder, or in n-Hexadecane, to select for plastic hydrolyzing microbes. Growth conditions, including temperature [31] and aeration [91], as well as the thickness of the PE film used as a substrate [50] all influence the efficiency of microbial degradation of PE.

### Limitations of the Methods and Techniques Used in Real Biodegradation Assays

Methods used in different biodegradation experiments are general directed to investigate one or more of the biodegradation stages. These test methods are insufficient in their ability to realistically predict the biodegradability in these environments, due to several shortcomings in experimental procedures and a paucity of information in the scientific literature [21]. So, generally, several experiments are used together.

One method used to access transformations in first stage of biodegradation and biodeterioration, is Fourier transform infrared spectroscopy (FT-IR) spectrometry, which has been the most commonly used method for the analysis of biodeterioration of polyethylene by photoxidation or heat. FT-IR techniques have been used to detect oxidative products of physical plastic degradation, such as the appearance of carbonyl-groups after UV-irradiation, treatment with oxidizing agents, or oxidizing enzyme from microorganisms as it a useful tool to determine the formation of new biodeterioration or disappearance of functional groups during biodegradation. In some reports, the deterioration of polyethylene was measured by the formation of carbonyl group, so degradation products, chemical moieties incorporated into the polymer molecules such as branches, co-monomers, unsaturation and presence of additives such as antioxidants can be determined by this technique [2,26,29,30,36,39,50,53,56,57,65]. It should be noted that biodeterioration is the first step in biodegradation and must not be reported as complete biodegradation.

Scanning electron microscopy (SEM) may provide evidence of physical deterioration (degradation and/or erosion) of the polymer surface where it has been colonized by the microorganisms or just show typical pattern of bacterial growth on the polymer surface [56]. According to Harshvardhan and Jha’s procedure [2], to survey the colonization of plastic materials, samples are removed from the culture medium and washed in phosphate buffer (pH 7.2) to release excess medium. In contrast, in the procedure for the examination of surface erosion, PE samples are washed with a 2% SDS solution in water followed by several rinses in warm distilled water to remove surface-adhered cells completely. Both types of PE samples are fixed in 2% glutaraldehyde in phosphate buffer (pH 7.2) for 2 h and dehydrated in graded ethanol (50%, 70%, and 100%). After fixation, the samples are dried in a vacuum. The dehydrated samples were sputter-coated usually with gold, which resulted in a thick gold layer. The samples are then examined using an Environmental SEM [2].

However, SEM images may show surface corrosion of polymers, SEM images alone do not provide evidence for complete biodegradation of polyethylene polymers. Examples of SEM images suggesting microbial degradation of LDPE are shown in Figure 4. Figure 4a,b shows polyethylene particles (particle size of 400 μm or less) and PE films, respectively, that have used by researchers in biodegradation experiments. Figure 4c,e, respectively represent images of colonization and attachment of *Sphigobacterium moltivorum* and *Pseudomonas putida* on the surface of PE particles (with different magnifications), while Figure 4d,f shows evidence of corrosion/penetration the PE by *Sphigobacterium moltivorum* and *Delftia tsuruhatensis* [37,66].

The simplest used way to monitor biodegradation is the measurement of gravimetric weight loss using a sufficiently sensitive scale [2,34,41,43]. To accurately determine the dry mass of residual polymers after a biodegradation experiment, PE material is removed from media. In case of polyethylene powder, PE with bound cells are filtered using a suitable filter paper. The filter pores must be small enough to capture the PE particles, yet wide enough for the mineral components and cells to be washed through. The filtered PE particles/PE film are then washed with 2% (*m*/*v*) sodium dodecyl sulfate (SDS) to lyse any remaining cells that adhered to the PE surface. The PE particles (on the filter paper)/PE film was further rinsed with distilled water and then dried overnight at 60 °C before weighing. Although some dried cells debris remained attached to the PE particles (visualized by scanning electron microscopy (SEM)), their contribution to the total PE particle mass can be considered negligible. However, the remaining cells may be removed by subjection the PE samples to ultra-sonication [37]. The rate of biodegradation is usually reported as the percentage of polymer weight loss per unit time of experiment as below [53,66];
% weight loss = [(initial weight − final weight)/initial weight]

However, weight loss can result from consumption of low molar mass PE material and may not be a good indicator of complete biodegradation of intact to fragmented polymer chains. Also, in the case of polyethylene powder, the recovery of particles from media is not accurate, as the particles maybe easily missed, and this measurement suffers from wide-range of standard deviation in data collection.

Some researchers have used gel permeation chromatography (GPC) to measure the change in molar mass and molecular mass number followed by microbial treatment [18,45]. Measurement the M_w_ and M_n_ can be considered as reliable methods to distinguish true biodegradation of intact to fragmented polyethylene, as increases or decreases in M_w_ and M_n_ reflect consumption of low molar mass chains versus high molar mass chains, respectively. For example, Reddy et al. [64] showed a small increase in the M_w_ of LDPE after treatment with a montmorillonite clay and *P. aeruginosa*, suggesting that this bacterium was able to degrade low molar mass compounds (presumably the chain-ends), but unable to hydrolyze the high molar mass fractions of the polymer.

Gas chromatography (GC) using non-polar columns has been used to determine biofragmentation and existence of saturated linear alkanes in culture media after microbial degradation of polyethylene [66]. Abrusci et al. (2011) and Kyaw et al. (2012) used GC-MS to detect saturated alkanes generated by microbial degradation of polyethylene [31,53]. The other techniques used to detect biodegradation include, measurement of dry biomass weight on polyethylene containing media (assimilation stage) [65], Epifluorescence microscopy to monitor the initial biofilm formation [56], size exclusion chromatography (SEC) to estimate changes in molar mass distribution [57], and carbon dioxide (CO_2_) measurement (mineralization) that shows the end of biodegradation [31,46].

## 8. Guidelines for Studying Microbial Degradation of Polyethylene

Generally, assays for microbial degradation of PE have been conducted by two different sets of researchers. Groups of researchers who have investigated PE degradation are generally environmentalists, using bulk PE materials of different types (LMWPE, LLDPE, LDPE, or HDPE), in natural environments, like soil, compost, or aquatic systems, with mixed, undefined populations of microorganisms, without attention to microbial type. Changes in appearance, weight loss, or mechanical properties of the PE are measured, and any change observed is called “biodegradation”. However, this approach is largely “trial and error”, and the factors affecting changes in the PE are not clear. Authors of these papers misunderstand the difference between deterioration and partial degradation. On the other hand, the advantage of these experiments is that they are conducted in the environment under ambient conditions, and the results reflect the reality of PE degradation. The other groups of researchers who have investigated PE degradation are microbiologists who are interested in the ultimate transformation of PE to CO_2_ and biomass (mineralization via true biodegradation). The biodegradation experiments are conducted with defined species of microorganisms from collections, or isolated using special media. Generally, all aspects of the experiments are well defined, and the authors understand the process of biodegradation. The application of molecular biology and genome sciences has begun to identify the specific genes and gene products involved in polyethylene degradation in this aspect.

The biodegradation of PE is a complex process that is influenced by many factors. A PE polymer chain can be exposed to different types of processes from production to treatment and sample preparation that change the polymer properties before microbial treatment. A wide range of microorganisms with different behavior and secreted substances make the biodegradation process complex and questionable. However, biodegradation experiments of polyethylene can be studied from both microbial and chemical aspects.

From a chemical point of view, the method of polymerization determines the type and amount of impurities, such as fragments and oligomers, in PE materials. There are significant variations in chain arrangement and chemical structure of PE, number and length of side chains (polymer architecture or topology), and crystallinity (physical statue). In addition, the chemical, and even the molecular structure, can be affected by the solvents used to solubilize PE powder. In other words, the solvent can become an incidental impurity in the enter PE structure. Functional groups created during polymerization or treatment process can influence both hydrophobicity as well as the accessibility of hydrolytic and/or oxidative enzymes to these functional groups. The type of PE used (film or powder) in biodegradation experiments can influence the surface contact area for microorganisms. Additives, antioxidants, and/or pretreatments by heat, UV-irradiation, and/or oxidative chemicals can change the microbial adsorption.

From a microbial point of view, microorganisms cannot degrade the polyethylene unless they are in direct contact with surface of polymer (in form of powder or film). This is not easily achieved, as PE is a hydrophobic material and not soluble in aqueous media. Some microorganisms can colonize the surface and may be able to access the polymer chain as carbon source. However, colonization and biofilm formation are not direct evidence for the microbial degradation of polyethylene. As mentioned in Section 6.3, biosurfactant production by bacteria is the key factor in biodegradation experiments and PE degradation. The secretion of biosurfactant molecules reduces surface tension and facilitates attachment of bacteria to the surface of PE. Thus, biosurfactant production must also be considered and measured by researchers in biodegradation experiments.

In culture media used in PE biodegradation experiments, there are often other carbon sources, in addition to PE polymers such as surfactants (Tweens, EDTA, etc), carbon sources from mineral salts as essential element for bacterial growth, impurities derived from the PE synthesis (oligomeres, monomers, functional groups,etc), impurities derived from sample preparation or treatment (solvents, free functional groups, etc), and substrates used to “preadapt” the bacteria, such as paraffin or n-hexadecane, (C16) that the microorganisms consume before being exposed to PE as a substrate.

Microorganisms consume and metabolize carbon sources in order to support their accessibility and molar mass. The border limit of transfer of molecules through bacterial cell wall is around 500 g/mol (35 carbons) (depends on the bacterial species and molecule shape). In the case of n-alkanes, it is reported that paraffin (44 carbons, 618 g/mol) can be assimilated directly by some bacteria [58]. In experiments with PE polymers with a M_w_ of less than 5000 g/mol, a large proportion of the fragments in the range of 100–2000 g/mol, are rapidly metabolized by bacteria [69]. As the border limit of material penetration through cell wall is fairly constant, secretion of PE-degrading enzymes is more important than the length of the polymer chain. The type and amount of enzyme production from bacteria is a key factor to access high molar mass polymer chain as the secondary carbon source in media in biodegradation experiment. As Peixoto et al. [8] showed, there is evidence of biodegradation of LDPE of 191,000 g/mol without pretreatment in the form of a decrease in crystalinity percentage. It has been suggested any PE fibers used for biodegradation experiments be branched and amorphous to be more accessible for microbial attack.

Some polyethylene-degrading bacteria require the PE to be treated by UV-radiation or other pretreatments before the bacterial enzymes can attack and degrade the polymer. Other bacteria, however, secrete enzymes that can initiate biodeterioration of PE polymers on their own. Compounds incorporated in the PE chains, such as functional groups and vinylydenes, are consumed in the next stage. Under the best conditions, tertiary carbons, chain-ends, and branches are attacked and separated from backbone in this stage of degradation. It is important how the polymer chain is amorphous or crystalline and how much is branched. In the final stage, after the more accessible segments attacked by enzyme and dispersed as low molar mass pieces in media (usually proved by GC-MASS analysis), the PE degradation process stops and the polymer backbone remains unchanged. Another important point that is PE-degrading bacteria essentially do not assimilate all the degradation products. Bacterial enzymes may cleave the polymer chain to fragments that exceed the border limit of wall penetration. Thus, the measurement of carbon dioxide as an indicator of assimilation of PE hydrolysis products alone is not a good indicator of PE degradation.

According to above, the following protocol is recommended (Figure 5): Sample preparation: PE samples can be in form of particles or films. In case of particles, the surface area for bacterial colonization is greater. Disadvantages of particle use include a) measurement of weight loss, which is prone to errors, and b) SEM analysis for assessment the penetration of microorganisms, which is impossible because of uneven surface. If any pretreatments are used, these should be done before bacterial treatment;Removing impurities: Impurities can be removed easily by washing the sample, and subsequently growing the common bacteria on culture media containing PE sample to remove the usual impurities added in previous steps through consumption;Measuring molar mass, molecular dispersity index and functional groups; Measurements of molar mass and molar mass number are still the best biodegradation assays. These must be used both immediately before and after of microbial degradation. An increase in molar mass after incubation with bacteria suggests that the bacteria consumed branches and other low molar mass portions of the polymers. An increase in the molecular dispersity index indicates that chain breakage occurred at the ends of the polymer chains or branches rather than at the center of the molecule. FTIR analysis can show changes in functional groups in the polymer structure.Bacterial treatment: In biodegradation experiments, it is assumed that bacteria able to degrade PE (by testing the growth on n-hexadecane or paraffin [67] and forming colony on polyethylene surface [31]). Bacterial treatment should be performed under optimum conditions for the microbe(s) used in the experiments. Bacteria are inoculated on culture media containing PE samples for further experiment.Establish microbial growth curves on polyethylene: It is essential to establish growth curves of the microorganisms used in biodegradation assays When comparing the biodegradation of PE between two bacteria, it is important to have accurate growth curves of the bacteria.

## 9. Conclusions and Perspectives

The process of biodegradation of PE is defined by four stages: biodeterioration, biofragmentation, bioassimilation, and mineralization. Complete biodegradation of PE requires a reduction in the polymer molar mass and molecular mass number as a consequence of fragmentation into smaller molecules that are subsequently catabolized by microorganisms. However, most studies of putative biodegradation of PE by microorganisms report biodeterioration, and few report biofragmentation. Further, evidence for bioassimilation and mineralization is lacking. Analyses of the genes and gene products that oxidize the alkane chains of polyethylene may lead to greater understanding of the molecular mechanisms of polyethylene biodegradation.

Complete PE biodegradation may be defined as the oxidation of intact polymers to highly fragmented polymers, with corresponding decreases in M_w_ and M_n,_ followed by subsequent conversion of the polymer fragments to CO_2_ and biomass (under aerobic conditions). Given the massive accumulation of PE in the environment, it is urgent that we understand the mechanisms of PE degradation by microorganisms. The most significant barriers to achieving this goal are the lack of a definition of true biodegradation of PE, and a general misunderstanding of the biodegradation process. A key weakness of the term ‘biodegradable’ is that it does not contain any information about the location, timescale, and extent of the decomposition process [67]. Standardized protocols for the microbial degradation of PE are required.

## Figures and Tables

**Figure 1 polymers-12-00123-f001:**
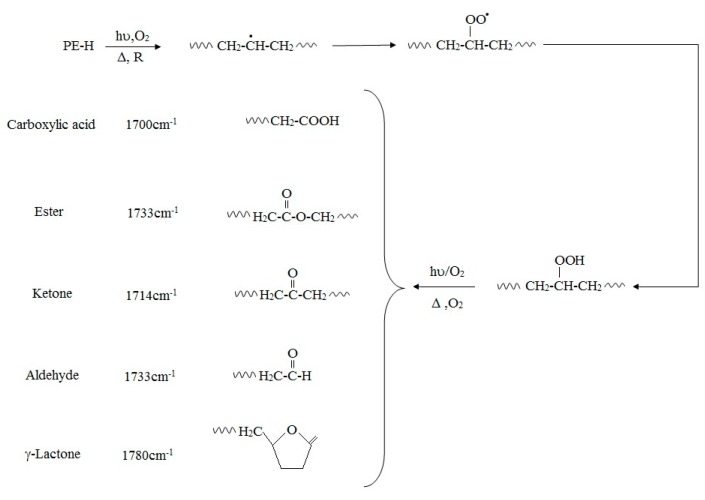
Degradation pathways of polyethylene containing pro-oxidant additives.

**Figure 2 polymers-12-00123-f002:**
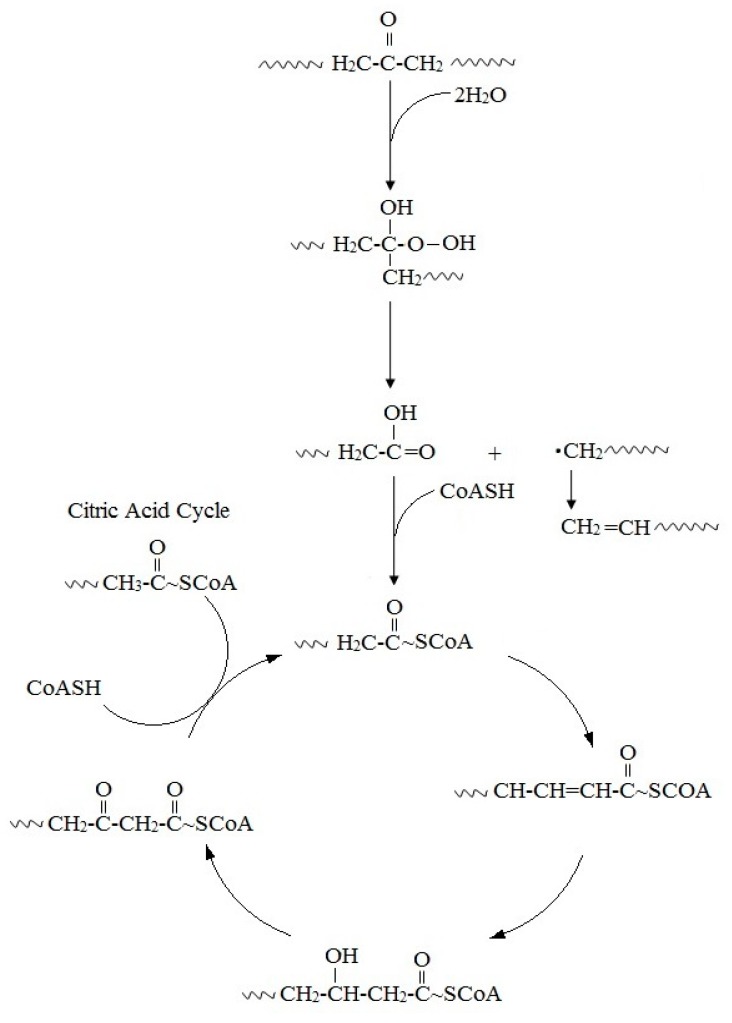
Proposed mechanism for the biodegradation of PE.

**Figure 3 polymers-12-00123-f003:**
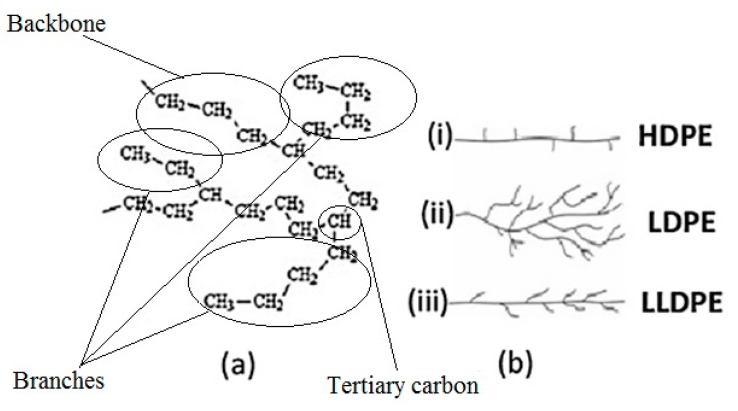
Polyethylene structure. (**a**) General chemical structure; (**b**) Schematic differences between (i) High-density Polyethylene (HDPE), (ii) Low-density Polyethylene (LDPE), and (iii) Linear Low-density Polyethylene (LLDPE) [from [18] with permission from the author].

**Figure 4 polymers-12-00123-f004:**
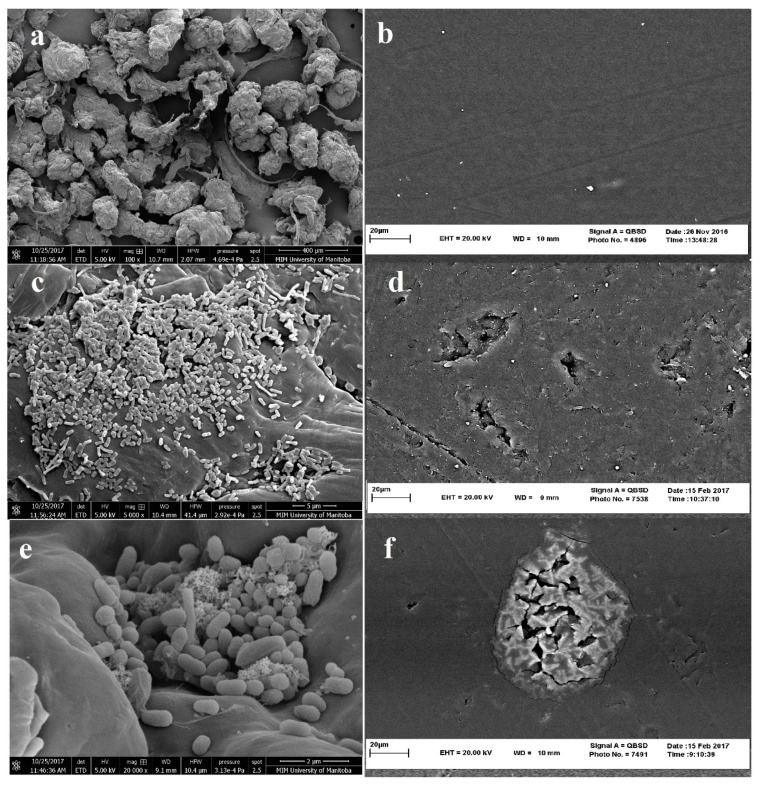
Scanning electron microscope image of (**a**) LDPE particles before treatment (magnification ×100); (**b**) LDPE film before treatment (magnification ×2000); (**c**) microbial colonization on PE particles by *Sphingobacterium moltivorum* (magnification ×5000); (**d**) Holes and penetration in PE sheet after treatment with *Delftia tsuruhatensis* (magnification ×2000); (**e**) microbial colonization by *Pseudomonas Putida LS46* (magnification ×20,000); and (**f**) Corrosion of PE sheet after treatment with *Sphingobacterium moltivorum* (magnification ×2000) (preparation of images by MIM unit at University of Manitoba and Central Laboratory at Ferdowsi University of Mashhad).

**Figure 5 polymers-12-00123-f005:**
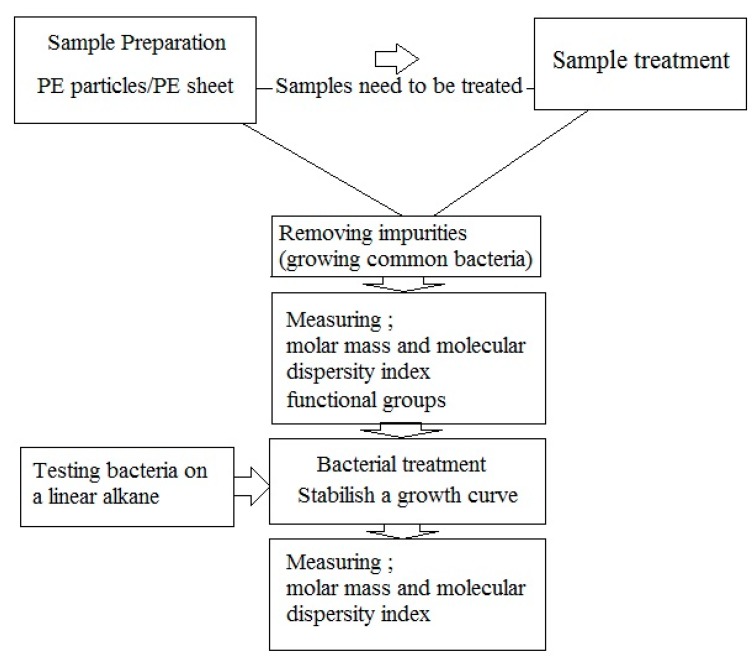
A suggested flow chart for biodegradation experiments.

**Table 1 polymers-12-00123-t001:** Published review articles on plastic biodegradation.

Authors	Year of Publication	Topic	References
Shimao	2001	Biodegradation of plastics	[9]
Koutny et al.	2006	Biodegradation of polyethylene films with prooxidant additives	[10]
Arutchelvi et al.	2008	Biodegradation of polyethylene and polypropylene	[11]
Shah et al.	2008	Biological degradation of plastics	[12]
Lucas et al.	2008	Polymer biodegradation: Mechanisms and estimation techniques	[13]
Tokiwa et al.	2009	Biodegradability of Plastics	[14]
Sivan	2011	New perspectives in plastic biodegradation	[15]
Ammala et al.	2011	An overview of degradable and biodegradable polyolefin	[16]
Restrepo-Flórez et al.	2014	Microbial degradation and deterioration of polyethylene	[17]
Sen and Raut	2015	Microbial degradation of low density polyethylene	[18]
Raziyafathima et al.	2016	Microbial Degradation of Plastic Waste: A Review	[19]
Emadian et al.	2017	Biodegradation of bioplastics in natural environments	[20]
Harrison et al.	2018	Biodegradability standards for carrier bags and plastic films in aquatic environments: A critical review	[21]

**Table 2 polymers-12-00123-t002:** Bacteria used in biodegradation studies of polyethylene (PE) degradation. The bacteria are listed alphabetically by genus.

Genus (and Species)	Source	Experiment Duration	Experiment Condition	Biodegradation Result	Reference
*Acinetobacter bumannii*	Municipal landfill	30 days	37 °C Non-pretreated PE	Biomass production	[42]
*Arthobacter defluvii*	Dumped soil area	1 month	PE bags	20%–30% W.L. *	[48]
*Bacillus amyloliquefaciens* *Bacillussubtilis*					
*Bacillus pumilus* *Bacillus subtillis*	Pelagic waters	30 days	PE bags	1.5%–1.75% W.L.	[2]
*Bacillus ssp.*	Waste coal, a forest and an extinct volcano crater	225 days	Modified PE	Reduction of mechanical properties by 98% No W.L. detected	[29]
*Bacillus sphericus*	Shallow waters of ocean	1 year	HDPE and LDPE; Untreated and Heat treated	3.5% and 10% 9% and 19%	[43]
*Bacillus megaterium* *Bacillus subtilis* *Bacillus cereus (MIX together)*	Soil	90 days	45 °C photo-degraded oxobiodegradable PE	7%–10% mineralization	[31]
*Bacillus amyloliquefaciens*	Solid waste dumped	60 days	LDPE	11%–16%	[49]
*Bacillus subtilis*	MCC No. 2183	30 days	Adding Biosurfactant Unpretreated 18 μm thickness PE	9.26% W.L.	[50]
*Bacillus pumilus M27* *Bacillus subtilis H1584*	Pelagic waters	30 days	PE bags	1.5–1.75 W.L. %	[2]
*Brevibacillus borstelensis*	DSMZ	90 days	50 °C Irradiated LDPE	17% W.L.	[51]
*Brevibacillus*	Waste disposal site	3 weeks	Pretreated PE	37.5% W.L.	[41]
*Chryseobacterium gleum*	Waste water activated sludge soil	1 month	UV-radiated LLDPE	-	[44]
*Comamonas sp.*	Plastic debris in soil	90 days	Non-treated LDPE	Changing in chemical properties	[8]
*Delftia sp.*	Plastic debris in soil	90 days	Non-treated LDPE	Changing in chemical properties	[8]
*Kocuria palustris M16,*	Pelagic waters	30 days	PE bags	1%	[2]
*Microbacterium paraoxydans*	Having Gene bank ID	2 months	Pretreated LDPE	61% W.L.	[52]
*Pseudomonas sp.*	Mangrove soil	1 month	PE	20.54% W.L.	[30]
*Pseudomonas aeroginosa*	Petroleum contaminated beach soil	80 days	LMWPE	40.8% W.L.	[45]
*Pseudomonas sp.*	Beach soil contaminated with crude oil	80 days	37 °C LMWPE	4.9%–28.6% CO_2_ production	[46]
*Pseudomonas sp.*	Garbage soil	6 months	PE bags	37.09% W.L.	[34]
*Pseudomonas citronellolis*	Municipal Landfill	4 days	LDPE	17.8% W.L.	[38]
*Pseudomonas sp.*	Having Gene bank ID	2 months	Pretreated LDPE	50.5% W.L.	[52]
*Pseudomonas aeroginosa* *Pseudomonas putida* *Pseudomonas siringae*	ATCC	120 days	Untreated PE	9%–20%	[53]
*Pseudomonas sp.*	Waste disposal site	3 weeks	Pretreated PE	40.5% W.L.	[41]
*Rhodococcus ruber*	PE agricultural waste in soil	4 weeks	Treated LDPE	Up to 8% W.L.	[36]
*Rhodococcus ruber*	PE agricultural waste in soil	60 days	LDPE	0.86% W.L./week	[54]
*Rhodococcus ruber*	PE agricultural waste in soil	30 days	LDPE	1.5%–2.5% W.L. Reduction of 20%.in Mw and 15%.in Mn	[55]
*Rhodococcus rhorocuros*	ATCC	6 months	27 °C Degradable PE	60% mineralization	[56]
*Rhodococcus rhorocuros*	ATCC 29672	6 month	PE containing prooxidant additives	Different amount of mineralization	[57]
*Rhodococcus sp.*	Waste disposal site	3 weeks	Pretreated PE	33% W.L.	[41]
*Rhodococcus sp.*	Three forest soil	30 days	LDPE containing prooxidant additives	Confirmation of Adhering	[35]
*Staphylococcus arlettae*	Various soil environments	30 days	PE	13.6% W.L.	[32]
*Stentrophomonas sp.*	Plastic debris in soil	90 days	Non-treated LDPE	Changing in chemical properties	[8]
*Stentrophomonas pavanii*	Solid waste dump site	56 days	Modified LDPE	Confirmed by FTIR	[40]
*Streptomyces spp.*	Nile River Delta	1 month	30 °C Heat treated degradable PE bags	3 species showed slight W.L.	[58]

* W.L., Weight loss report as %.

**Table 3 polymers-12-00123-t003:** Characterization peaks in FT-IR [50].

SI No.	Wave Number (cm^−1^)	Bond	Functional Group
1	3000–2850	–C–H stretch	Alkanes
2	2830–2695	H–C = O: C–H stretch	Aldehyde
3	1710–1665	–C = O stretch	Ketones, Aldehyde
4	1470–1450	–C–H Bend	Alkanes
5	1320–1000	–C–O stretch	Alcohol, Carboxylic acid, esters, ethers
6	1000–650	=C–H Bond	Alkenes

**Table 4 polymers-12-00123-t004:** Fungi capable of PE biodegradation.

Species	Reference
*Mucor rouxii; Aspergillus flavus*	[58]
*Penicillium simplicissimum*	[65]
*Cladosporium cladosporoides; Nocardia asteroides*	[56]
*Fusarium sp.*	[33]
*Aspergilus sp.; Fusarium sp*	[87]
*Gliocladium viride, Aspergillus awamori, and Mortierella subtilissima*	[29]
*Aspergilus sp.*	[88]
*Zalerion maritimum*	[89]

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
