# Peer review of "Challenges with Verifying Microbial Degradation of Polyethylene"

_polymers, 2020, doi:10.3390/polym12010123_

Round 1

Reviewer 1 Report

Your paper  is touching a very significant area  that, in my judgement, has not been researched extensively  in the way it deserves. I personally enjoyed it. However there is a significant part missing  that I really like to see:

As you may be aware, in many industries, they are of the opinion  that if for their underground fire water rings , they replace carbon steel with non-metals (mainly HDPE), they will be immune to biodegrdation, particularly by bacteria such as but not limited to anaerobic sulphate reducing bacteria (SRB).

I would like to see a section only on  biodegradation of HDPE and if possible by bacteria such as SRB  and suphur oxidising bacteria.

Once again, I do congratulate to you  for this beautiful piece of work.

Dr. Reza Javaherdashti

Author Response

Comment: I would like to see a section only on biodegradation of HDPE and if possible by bacteria such as SRB and suphur oxidising bacteria.

Response: A section on “Biodegradation of HDPE” was added at pages 18 and 19. Also a reference was added in references list (reference [73]). However, we feel that details about biodegradation by bacteria such as SRB and sulphur oxidising bacteria is beyond the scope of this paper, give the time limit required by the Editor. In other words, we would require more time to add information about biodegradation by sulphur reducing- and sulphur oxidising- bacteria

Reviewer 2 Report

The paper “A review Challenges with Verifying Microbial Degradation of Polyethylene” is a interesting review on biodegradation of polyethylene. It is an interesting subject and the authors summarize an accurate description and comparison of microbial degradation experiments.

I suggest to expand and include a more detailed explanation and references in the section 7.1 (SEM technique, gravimetric weight).

In summary, I would recommend this work for publication in Polymers after minor revisions.

Author Response

Comment: I suggest to expand and include a more detailed explanation and references in the section 7.1 (SEM technique, gravimetric weight).

Response: Some notes about FTIR, and a new Table (Table 3) were added on page 12 . A paragraph on the use of SEM and procedures was added on page 31, and a Figure with SEM images was added on page 32. Paragraphs with more explanation of weight loss measurement and sample preparation were added on page 33.

Author Response

Comment: I suggest to add a figure showing the chemical changes in structure during the biodeterioration as well as the biodegradation, i.e. highlighting the oxidative processes, functional groups formed and the further transformed etc as well as the more architectural highlights during degradation differentiating between side chains and polymer backbone.

Response: Two figures (Figures 1 and 2) were added to the text at pages 11 and 15, respectively. highlighting the oxidative processes, functional groups formed and the further transformed productions. Also, some more details in PE chain including backbone, tertiary carbon and branches were inserted in Figure 3.

Comment: I recommend to include a figure depicting a suggested protocol: what should be done in sample preparation, which properties should be characterized before starting the degradation experiment (and at what time point in the process) etc.

Response: Figure 5, shows simple protocol of biodegradation on page 39.

Comment: Not necessary, but perhaps nice to the eye would be a collection of pictures from cited articles showing the degradation, e.g. holes in PE, change of surface etc.

Response: Figure 4 has been added to prepare SEM images of PE particles, PE films, and schematic of surface changing of polymer after biodegradation.

Comment: Such a suggested protocol in words would be really good also in the conclusion part. I believe that this would tremendously increase the impact of the work.

Response: The suggested protocol and other useful notes were added in Section 8, now titled “Guidelines for studying microbial degradation of polyethylene”.  

Scientific concepts/wording

Comment: It should be weight average molar mass (Mw) and number average molar mass (Mn) [subscripted w/n, and not “molecular weight”].

Response: The letters “n” and “w” in Mw and Mn ave been formatted as subscript throughout the manuscript. “Weight-averaged Molecular weight” and “Number-average Molecular weight” have changed to “Weight-average molar mass” and “Number-average molar mass”.

Comment: Also in other parts, molecular-weight is used (it should be molar mass), and even “molecular number” / “molecular mass number” are used (which are not defined by IUPAC at all).

Response: “molecular-weight” was corrected as “molar mass” throughout the document and “molecular number” was changed to “molecular mass number”.

Comment: Be aware that molar mass should always be given with the unit g/mol (or kg/mol), but not by (k)Da anymore. These used to be applied synonymously, but have by now be defined differently. Also when reporting older studies that at that time have used Da, this should probably be updated here, and it would be good to use one term correctly and consistently throughout the text.

Response: All mentioned molar mass values were given the unit g/mol instead of Da/KDa or Dalton.

Minor points

Comment: Title: “a review Challenges….” (“a review” can be deleted IMO)

Response: “A review” was deleted.

Comment: Table 1: Format: caption needs to be moved into an own line. Table itself: remove country column, as it is not relevant

Response: Country column was removed.

Comment: Line 75: “(usually 150 rpm)” – is this really relevant? If it is, it should be explained completely, i.e. giving the shaking speed + other details, e.g. CO2 content etc.

Response: “usually 150 rpm” was removed.

Comment: Table 2: In some instances, only a % of change is given without referring to a property (e.g.W.L.) Either the W.L. was forgotten here, or some other property was used, in any case, please add this information.

Response: Cases without any referring to W.L or etc, indicates the authors have not stated anything about that information. All information that was accessible in the papers was cited exactly according to author’s explanations.

Comment: Lines 227‐243. Please add a sentence evaluating the used low molar masses of PEs in the published studies. Do you recommend that also for further studies? (Altogether, not only list the approaches of the different studies, but add an evaluation from your point of view).

Response: The related response was inserted in conclusion section.

Comment: Lines 269‐277: is the blending also leading to lower crystallinity, which may increase the rate of degradation?

Response: The properties of final blend depend on properties of every component added to the blend. For example percentage crystallinity decreased as the starch content increased in LLDPE/starch blends that may resulted in an increase of biodegradation.

Comment: Lines 295 + 305 (perhaps also in other parts). I would use the term “backbone” instead of “core”, as for me the core of a polymer could also relate to the inner part of a fiber rather than the molecular core.

Response: In related section, all words of “core” replace with backbone.

Comment: Line 298ff: rephrase ‐ PE is generally not cross-linked (in terms of covalent bonds), and it is not held together by hydrogen bonds (which require polar donor and acceptor groups such as –OH, ‐NH2, ‐COOH etc), but more likely by van‐der‐Waals interactions.

Response: In this case, crosslinked PE means the linear carbon chains with accessible short side-chains (branches). The correct changes were made in main text according to reviewer comment.

Comment: Line 556: the chemical structure should more refer to the “chain arrangement” (you probably refer to the polymer architecture or topology, i.e. linear, low or high branched, length of side chains”) than to Mw, Mn and the polydispersity.

Response: The paragragh was corrected as “There are significant variations in chain arrangement and chemical structure of Polyethylene like linear, low or high branched, length of side chains (polymer architecture or topology) and cristallinity (physical statue)”

Typos etc.

Comment: Line 17: “suffer from a number of limitations” or “suffer from numerous limitations”

Response: “from a number limitation” changed to “a number of limitations”

Comment: Line 22: involved in microbial

Response: “involved microbial” changed to “: involved in microbial”

Comment: Line 26: simplify

Response: “simply” changed to “simplify”

Comment: Line 32: Polyvinyl chloride……terephthalate….

Response: “polyvinyl chloride” and “terphetalate” changed to “Polyvinyl chloride” and “terephthalate” respectively.

Comment: Line 84: The sentence should be reworded as “…meaningful comparisons….are not valid.” does not make sense to me; e.g. use “…comparisons …. are not valid” or “…comparisons….are not meaningful.”

Response: “meaningful comparisons of the various results of biodegradation may not be valid.” corrected as “comparisons of the various results of biodegradation are not meaningful.”

Comment: Line 101ff: “…of PE polymers mainly occurs through abiotic processes….” Then add at the end of line 103: “(but see also chapter 3 – biodeterioration)”. The reason is that otherwise the text seemingly contradicts itself.

Response: This comment does not make sense to us. There does not seem to be a contradiction between lines 101 and 103, and the first paragraph of Section 3 “Biodeterioration”, in which we explain that some microorganisms can initiate “biodeterioration”.

Comment: Line 106: “The exposure of PE….”

Response: “that exposure of PE” changed to “The exposure of PE….”

Comment: Line 118f: delete: “, both in the IR”

Response: “, both in the IR” was deleted.

Comment: There are a couple of text passages and references (e.g. line 174, 186, 259 ff etc) that are underlined – probably false formatting. Please correct

Response: Underlines have been removed.

Comment: Line 234: delete most of the repetitive sentence starting with Yoon et al. Instead: “Such Pes were degraded….”

Response: The suggested changes were made and the sentences were corrected as “while Yoon et al. [46] tested the biodegradation of non-oxidized LMWPE of 1,700 to 23,700 Da [33, 41]. Such Polyethylenes was degraded by….”

Comment: Line 242: delete “also”

Response: “also” at line 242 was deleted.

Comment: Line 246: “, usually a solvent like…”

Response: “…usually chemical solvent like…” changed to “…usually a solvent like”

Comment: Line 249: rephrase “physical damage like liquid nitrogen” (isn’t a verb missing here?)

Response: The phrase was changed to “…the researcher uses solvents or liquid nitrogen to apply physical damage [24], in order to prepare…”

Comment: Line 262-266: split sentence in two sentences.

Response: The sentence was revised as suggested

Comment: Footnote 2: Degradable PE is also sometimes used when an polyester is produced with very few ester groups and properties similar to PE (Chem. Rev. 2016, 116, 4597−4641.)

Response: The reviewer’s comment was added to the footnote      

Comment: Line 267: “in order to accurately asses microbial…”

Response: “in order to accurate assay of microbial…” was corrected as “in order to accurately assess microbial….”.

Comment: Line 274: of the LDPE of more than

Response: “of” was added to the sentence.

Comment: Line 368: are the materials added to

Response: “are the material add to” was corrected to “are the material added to”

Comment: Line 447: 6.3 “Using bacteria that can form biofilms and secrete biosurfactans”

Response: The subheading title was changed according to the reviewer’s comment.

Comment: Line 470: also, not ALSO

Response: “ALSO” was corrected.

Comment: Line 491: One method used

Response: “One of method used” was changed to “One method used”

Comment: Line 492: Fourier transform infrared spectroscopy

Response: “Fournier Transformation-Infra Red” changed to “Fournier Transformation-Infra Red”

Comment: Line 506: the result from consumption

Response: “of” changed to “from”